# Effect of Dapagliflozin on Exercise Capacity and Cardiovascular Risk in Patients with Heart Failure

**DOI:** 10.3390/healthcare10112133

**Published:** 2022-10-26

**Authors:** Rakan Nazer, Mohammed Albratty, Monira I. Aldhahi, Maiasa Alqurashy, Maryam A. Halawi, Ali Albarrati

**Affiliations:** 1Cardiac Sciences Department, College of Medicine, King Saud University, P.O. Box 2460, Riyadh 11451, Saudi Arabia; 2Department of Pharmaceutical Chemistry & Pharmacognosy, College of Pharmacy, Jazan University, P.O. Box 114, Jazan 45142, Saudi Arabia; 3Department of Rehabilitation Sciences, College of Health and Rehabilitation Sciences, Princess Nourah Bint Abdulrahman University, P.O. Box 84428, Riyadh 11564, Saudi Arabia; 4Internal Medicine Department, Security Forces Hospital, P.O. Box 11481, Riyadh 11481, Saudi Arabia; 5Department of Pharmacy Practice, College of Pharmacy, Jazan University, P.O. Box 114, Jazan 45142, Saudi Arabia; 6Department of Haematology, Division of Cancer & Genetics, School of Medicine, Cardiff University, Cardiff CF14 4XN, Wales, UK; 7Rehabilitation Health Sciences Department, College of Applied Medical Sciences, King Saud University, P.O. Box 2460, Riyadh 11451, Saudi Arabia

**Keywords:** dapagliflozin, heart failure, type 2 diabetes mellitus, exercise capacity, cardiovascular risk

## Abstract

Heart failure (HF) is a serious disorder that affects millions of people worldwide, with a high rate of exercise intolerance, rehospitalization, and death. HF has many underlying causes, including type 2 diabetes mellitus (T2DM), which corresponds with high mortality and short survival among patients with HF. Numerous studies have shown the crucial role of gliflozins, a new generation of blood glucose-lowering medications, in cardiac remodeling, with beneficial impacts on exercise capacity and cardiovascular (CV) mortality, even in non-diabetic individuals. The foundational CV-protective frameworks of these agents are intricate and multifaceted. Dapagliflozin is a new widely used drug and a valuable alternative for patients with T2DM and CV risk factors. Dapagliflozin was approved by the Food and Drug Administration (FDA) in 2019 to lower the risk of HF hospitalization in patients with concurrent T2DM and CV disease or associated risk factors. However, the effects of this new drug on exercise capacity and CV risk still need to be elucidated. The primary objective of this review is to summarize the effect of dapagliflozin on exercise capacity and CV risk in patients with HF.

## 1. Introduction

Heart failure (HF) is a leading cause of mortality and morbidity worldwide [1]. HF is associated with several risk factors, the most prominent of which is T2DM, which has a high risk of mortality and short survival span. In addition, individuals with T2DM have a greater risk of HF independent of coronary heart disease, and some studies support HF prevention therapies [2,3,4]. These two abnormalities may be addressed effectively on their own, but establishing a treatment for HF associated with T2DM remains a challenge.

Sodium–glucose cotransporter-2 (SGLT-2) inhibitors are a class of hypoglycemic drug that was recently discovered to have potential cardiovascular-protective effects in adults [5,6]. SGLT-2 inhibitors modulate the sodium-glucose cotransporter in the proximal convoluted tubule of the kidney, preventing glucose reabsorption and lowering blood glucose by increasing urine glucose excretion [5]. SGLT-2 inhibitors have also been reported in recent studies to lower blood pressure and body weight, which may have cardiovascular implications [6,7,8,9,10,11]. One of the SGLT-2 inhibitors is dapagliflozin, which is commonly used in HF clinics. Several well-designed randomized controlled trials (RCTs) of dapagliflozin have shown remarkable cardiovascular benefits [7,8].

Exercise capacity is defined as the ability of the cardiovascular system to supply sufficient oxygen to the musculoskeletal system during exercise and extract the supplied oxygen from the blood by the exercising muscles [9]. Exercise capacity is usually reduced in patients with HF and is characterized by an inability to sustain physical effort [9]. Dapagliflozin has demonstrated cardiovascular safety and efficacy in reducing cardiovascular events and related hospitalizations associated with HF regardless of the incidence of T2DM [10,11,12]. However, there is a controversial effect on exercise capacity. Therefore, the objective of this narrative review is to highlight the effect of dapagliflozin on exercise capacity and cardiovascular risk in individuals with HF reported in clinical studies. We intended to critically review the results of the studies and summarize the effect of dapagliflozin on exercise capacity and cardiovascular risk.

### Dapagliflozin as a Drug

Dapagliflozin is a selective SGLT-2 inhibitor that reduces renal glucose absorption by inhibiting the SGLT-2 receptors present in the S1 region of the proximal kidney tubules [12]. Dapagliflozin was recently approved in the United States for lowering the risk of HF hospitalizations in patients with T2DM and cardiovascular disease risk markers based on these promising results [10]. Dapagliflozin works by altering visceral fat and blood glucose levels, which are both elements of metabolic syndrome and linked to the severity of cardiovascular disease. The physiological mechanisms by which dapagliflozin produces its cardioprotective benefits are uncertain, and more research is anticipated. Dapagliflozin is used as a monotherapy in patients with T2DM who are resistant to metformin or for whom it is not recommended. In terms of lowering the levels of hemoglobin A1c (HbA1c), dapagliflozin is comparable to metformin as monotherapy [13]. Furthermore, in patients with suboptimal glycemic control, dapagliflozin could be used to supplement existing antihyperglycemic drugs. Dapagliflozin has a half-life of 12.9 h and is excreted mostly in the urine as its 3-O-glucuronide metabolite [10]. There is greater systemic circulation of dapagliflozin in patients with renal and hepatic dysfunction [14].

Dapagliflozin protects the heart and circulatory system through a complex mechanism. Improvements in the circumstances of ventricular preload, cardiac metabolism and bioenergetics, Na+/H+ exchange, sugar and lipid metabolism, circulatory load, cardiovascular system, and other factors could all be part of the mechanistic strategy (Figure 1) [15]. However, the underlying mechanisms of the prevention of HF are thought to be a relief of ventricular loading conditions and reduction of preload through diuretic and natriuretic actions. Dapagliflozin has an effect on the functionality of cardiac cells and a putative effect on modulating the cardiac physiology in HF [16,17]. A number of cardioprotective sequelae of dapagliflozin contribute to altering HF pathophysiology, including diuresis/natriuresis, blood pressure reduction, erythropoiesis, enhanced cardiac energy metabolism, inflammatory minimization, suppression of the sympathetic nervous system, and prevention of adverse cardiac remodeling [18]. In vitro, dapagliflozin causes adenosine 5′-monophosphate activated protein kinase (AMPK) phosphorylation, resulting in down-regulation of protein kinase (PKC) phosphorylation in cardiac myoblast H9c2 cells after hypoxia/reoxygenation (H/R) treatment. The AMPK/PKC nicotinamide adenine dinucleotide phosphate hydrogen (NADPH) oxidase pathway has shown that dapagliflozin therapy reduces H/R-induced oxidative stress. Through AMPK/PKC/NADPH oxidase signaling, dapagliflozin also corrects H/R-induced abnormalities in PGC-1 expression, mitochondrial membrane potential, and mitochondrial DNA copy number. Dapagliflozin appears to have the ability to reduce ischemia/reperfusion (I/R) induced oxidative stress and, subsequently, cardiac apoptosis by modulating AMPK, minimizing the cardiac dysfunction mediated by I/R injury [19]. In Sprague Dawley rats, dapagliflozin has been shown to alleviate angiotensin II-induced cardiac remodeling by modulating transforming growth factor (TGF)-1/Smad signaling in a non-glucose-lowering-dependent mechanism [11]. Another study suggested that dapagliflozin significantly reduces the high glucose-induced endothelial–mesenchymal transition (EndMT) in human umbilical vein endothelial cells (HUVECs) and fibroblast collagen secretion. Similarly, dapagliflozin abolishes upregulation of TGF/Smad signaling and inhibition of AMPK activity. The anti-EndMT actions of dapagliflozin were then reversed in HUVECs using AMPK siRNA and compound C. In addition, dapagliflozin can protect against dilated cardiomyopathy and myocardial fibrosis by decreasing fibroblast activation and EndMT through suppression of AMPK-mediated TGF/Smad signaling [20]. In a mouse model with transverse aortic constriction, dapagliflozin treatment improved cardiac systolic performance and prevented myocardial fibrosis and cardiomyocyte death, suggesting that it could be used as a novel therapeutic to combat pathological cardiac remodeling in non-diabetics [21].

## 2. Methods

This narrative review was carried out to identify the effect of dapagliflozin on exercise capacity and cardiovascular risk in patients with HF in the presence and absence of T2DM. Key databases were used to identify relevant studies related to this topic, including PubMed, Web of Science, Ovid gateway, and clinical trial website. Key search terms were associated with dapagliflozin (i.e., “SGLT-2 inhibitor”, “sodium-glucose cotransporter 2 inhibitor”, “dapagliflozin”, “exercise capacity”, and “cardiovascular risk”) and HF. The literature searches were limited to clinical trials and meta-analyses of dapagliflozin published in English between January 2019 and September 2022. Appropriate publications identified through the literature search results were included in this review.

## 3. Findings

### 3.1. Effect of Dapagliflozin on Exercise Capacity

One of the defining symptoms of HF is reduced exercise capacity, which is linked to lower health-related quality of life and worse prognosis. Ample studies have investigated several pharmacological treatments to improve exercise capacity in patients with HF, but no drug can yet improve exercise capacity in this population. The 6 min walking test (6MWT) is a widely available tool for determining exercise capacity in patients with HF [22]. Although the cardiopulmonary exercise test is still the gold standard for measuring exercise capacity in patients with HF, the 6MWT may facilitate a more objective measure of the patient’s daily activities. The 6MWT can be used to assess a patient’s exercise capacity, particularly in patients with advanced disease and various comorbidities who are unable to perform more rigorous exercise evaluations, such as patients with HF [23]. The 6MWT has been studied for its predictive value in morbidity and mortality, and the examination has been used to measure the response to different therapeutic interventions in a variety of patient populations, including HF. A number of well-conducted RCTs have examined the effect of dapagliflozin on exercise capacity in patients with HF (Table 1).

The clinical study Dapagliflozin Effects on Biomarkers, Symptoms, and Functional Status in Patients with HF with Reduced Ejection Fraction (DEFINE-HF) was designed to test the hypothesis that therapies involving dapagliflozin improve health and functional status in patients with HF with reduced ejection fraction (HFrEF) in the presence or absence of T2DM. The main objective of the trial was to determine how dapagliflozin affects the specific health status of HF disease measured by the Kansas City Cardiomyopathy Questionnaire (KCCQ) and the exercise status measured by 6MWT [8]. For a 12-week period, 263 patients were randomized to placebo or dapagliflozin at a dose of 10 mg/d. After 12 weeks, the total KCCQ score significantly increased by 61.5% in the dapagliflozin group vs. 50.4% in the placebo group (*p* = 0.03). The 6MWT did not show significant differences between the patients treated with dapagliflozin and placebo (304 m vs. 301 m; *p* = 0.79). This discrepancy between the patient’s reported health status on the questionnaire and the patient’s performed test could be attributed to the fact that the 6MWT was more strongly reliant on the patient’s effort at a single moment in time, which could have led to significant variability. In addition, even if HF-related physiological limits were alleviated, comorbidities unconnected to HF, such as orthopedic constraints, could have impaired the 6MWT. 

Furthermore, preliminary results have been reported on the clinicaltrials.gov website for the Dapagliflozin Effect on Exercise Capacity Using a 6-min Walk Test in Patients with Heart Failure (DETERMINE) trials, which examined how dapagliflozin affects self-reported symptoms, functional disabilities, and walking distance in individuals with HFrEF (DETERMINE-Reduced; NCT03877237) or HF with preserved ejection fraction (HFpEF) (DETERMINE-Preserved; NCT03877224). In DETERMINE-Reduced [24], HFrEF patients had left ventricular ejection fraction (LVEF) ≤ 40%, NT-proBNP ≥ 400 pg/mL (≥300 pg/mL if hospitalized for HF in the past 12 months, ≥800 pg/mL if they had atrial fibrillation), and estimated glomerular filtration rate (eGFR) ≥ 25 mL/min/1.73 m^2^. In this trial, 313 patients with HFrEF were randomized to receive either dapagliflozin (10 mg once daily) or placebo for 16 weeks. In the dapagliflozin group, a significant improvement in clinical symptoms was measured by the KCCQ-symptom score. The median difference in KCCQ symptoms between the dapagliflozin and placebo groups was 4.23 (95% confidence interval [CI] 0.96 to 8.22; *p* = 0.02). In the dapagliflozin group, 35.3% of patients saw a significant improvement in HF symptoms (change in KCCQ-symptom score from baseline to week 16 = 15 points), compared to 24.5% in the placebo group (odds ratio, 1.6 [95% CI 1.0 to 2.7]). The median physical limitation score on the KCCQ was not significantly different between the dapagliflozin and placebo groups (median = 4.17; *p* = 0.05). However, the proportion of patients experiencing moderate to large improvement in physical limitations was greater in the dapagliflozin group (38.0% vs. 26.7% in placebo). This multicenter randomized clinical trial also showed that exercise capacity measured by the 6MWT was similar between the dapagliflozin group and placebo group (median difference between groups: 3.2 m [95% CI −6.5 to 13.0 m], *p* = 0.68) after 16 weeks of treatment. Furthermore, there were no significant differences in the proportion of patients with moderate or large improvement (≥31 m) in the 6MWT between the dapagliflozin group and the placebo group (41.3% vs. 36.5%, OR 1.1 [95% CI 0.7 to 1.8]). Despite the clear effects of dapagliflozin on symptoms in patients with HFrEF, the noted results appeared not to influence the 6MWT. 

A recent RCT looked at the effects of dapagliflozin on exercise capacity in 90 stable patients with HFrEF at 4 and 12 weeks [25]. Patients with HFrEF underwent incremental cycling cardiopulmonary exercise testing and the 6MWT. The baseline measurements of peak oxygen consumption (VO_2_) and 6MWT were 13.2 ± 3.5 mL/kg/min and 363 ± 10 m, respectively. Patients with HFrEF who were receiving dapagliflozin demonstrated a significant improvement in peak VO_2_ after 4 weeks, with a mean change of 1.09 mL/kg/min (95% CI 0.14–2.04; *p* = 0.02) and 1.06 mL/kg/min (95% CI 0.07–2.04; *p* = 0.03) after 12 weeks. However, there was no improvement in the 6MWT during the follow-up period. This was the first study to directly explore the effect of dapagliflozin on exercise capacity using the gold standard assessment in patients with HFrEF and find a beneficial effect. The lack of a positive effect on the 6WMT can be related to several factors, including a lack of encouragement and instructions given to patients, which led patients to not putting more effort in during the test and just walking at their own pace. Similarly, Kosiborod et al. [26] analyzed data from the Dapagliflozin and Prevention of Adverse Outcomes in Heart Failure (DAPA-HF) trial to examine the effects of dapagliflozin on symptoms, function, and quality of life in patients with HFrEF. The dapagliflozin group (10 mg/once daily) had a significant improvement in the mean KCCQ-symptom score, clinical score, and overall score at 4 months (1.9, 1.8, and 1.7 points) compared to the placebo group, respectively (all *p*-values < 0.0001). These improvements in the mean KCCQ symptom score, clinical score, and overall score continued over time and reached 2.8, 2.5, and 2.3 points, respectively, at 8 months (all *p*-values < 0.0001).

In contrast to patients with HFrEF, the PRESERVED-HF trial examined the effect of dapagliflozin in patients with HFpEF [27]. The dapagliflozin group (152 patients) showed significant improvements in the KCCQ-symptom score (5.8 points [95% CI 2.3–9.2]; *p*  =  0.001), physical limitation score (5.3 points [95% CI 0.7–10.0], *p*  =  0.026), and total score (5.8 points [95% CI 2.0–9.6], *p* = 0.003). The dapagliflozin group also exhibited improvement in the 6MWT at 12 weeks (effect size 20.1 m [95% CI 5.6–34.7]; *p*  =  0.007).

These findings were not replicated in the DETERMINE-Preserved trial [28]. In patients with HFpEF, dapagliflozin had no effect on the KCCQ-symptom score, KCCQ-physical level score, or the 6MWT. The median difference between dapagliflozin and placebo groups was 3.16 (95% CI 0.36 to 6.01; *p* = 0.07) in the KCCQ-symptom score, 3.12 (95% CI–0.09 to 5.37; *p* = 0.23) in the KCCQ-physical level score, and 1.6 m (95% CI −5.9 to 9.0; *p* = 0.66) in the 6MWT. Although these unpublished results did not show a significant effect on exercise capacity, we cannot exactly determine whether these results are realistic and representative because the complete picture of the involved sample is still unknown. 

### 3.2. Effect of Dapagliflozin on Cardiovascular Risk

According to emerging molecular findings, dapagliflozin may be effective in the treatment of HF regardless of EF or the presence of T2DM. A number of clinical trials have been conducted to evaluate the efficacy of dapagliflozin in the treatment of patients with HF, regardless of the presence of T2DM, to reduce the incidence of cardiovascular events (Table 2) [29,30,31,32,33,34]. 

The Dapagliflozin and Prevention of Adverse Outcomes in Heart Failure (DAPA-HF) study was designed to evaluate the efficacy and safety of dapagliflozin in patients with HFrEF, regardless of whether the enrolled patients had diabetes [29]. The DAPA-HF study randomized 4744 patients (1109 women) with HF and an LVEF ≤ 40% to placebo or dapagliflozin administered at a dose of 10 mg once daily in addition to standard care medications [29]. The composite occurrence of worsening HF or cardiovascular death was the primary outcome. The primary endpoint appeared in 16.3% of patients in the dapagliflozin group and 21.2% of patients in the placebo group over 18.2 months (*p* < 0.001). In the dapagliflozin group, 10.0% of patients experienced a first episode of worsening HF, compared to 13.7% in the placebo group. Cardiovascular death occurred in 9.6% of patients in the dapagliflozin group, whereas 11.5% in the placebo group died of any cause. Regardless of T2DM, dapagliflozin reduced the risk of HF deterioration and death from cardiovascular events compared to the placebo group. In terms of safety, 111 and 116 patients in the dapagliflozin and placebo group, respectively, discontinued the study due to adverse events (*p* = 0.79). Reduced cardiac volume, renal dysfunction, fractures, amputations, major episodes of hypoglycemia, and ketoacidosis were the observed adverse events that occurred with similar regularity in both groups. The DAPA-HF study presented a significant advancement in the treatment of HF with reduced EF, as its findings suggest that dapagliflozin could be incorporated as a fourth kind of therapeutic agent into the conventional therapy regimen, making dapagliflozin a first-in-class drug. 

Dapagliflozin decreases the occurrence of episodes of worsening HF or cardiovascular death equally and safely in both men and women to the same level as placebo (hazard ratio [HR] 0.73 and 0.79, respectively; *p* for interaction = 0.67) [31]. The DAPA-HF study showed that dapagliflozin reduces outpatient episodes of HF worsening, which are prevalent in patients with HF and are of prognostic significance [32]. The extrapolated mean event-free survival for individuals aged 65 years from a primary composite endpoint event was 6.2 vs. 8.3 years for placebo vs. dapagliflozin, representing an event-free survival time gain of 2.1 years. While considering death from any cause, the mean extrapolated life expectancy was noted to be 9.1 years and 10.8 years for placebo compared to dapagliflozin, representing an event-free survival time gain of 1.7 years. These findings suggest that dapagliflozin improves the projected event-free and overall survival in clinically relevant ways [32]. In a nested study from the Dapagliflozin Effect on Cardiovascular Events–Thrombolysis in Myocardial Infarction 58 (DECLARE–TIMI 58) trial, the effect of dapagliflozin on cardiovascular events was evaluated in 1724 patients with HFrEF [33]. The patients with HFrEF were randomized to placebo or dapagliflozin (10 mg daily) in addition to the standard care and followed for a median of 4.2 years. Major adverse cardiovascular events (MACEs) and a summary of cardiovascular death or hospitalization related to HF were the primary efficacy endpoints. Dapagliflozin did not reduce the incidence of MACEs (8.8% vs. 9.4% in placebo group; *p* = 0.17). However, dapagliflozin did lower the rate of cardiovascular death related to HF (HR 0.79 [95% CI 0.66–0.99]) and hospitalization (HR 0.64 [95% CI 0.49–0.95]). Other clinical trials on the effect of dapagliflozin in patients with HF, including the DETERMINE, DEFINE-HF, and PRESERVED, have not evaluated MACEs, cardiovascular death, or hospitalization related to HF in this population.

Furthermore, the Dapagliflozin in Heart Failure with Mildly Reduced or Preserved Ejection Fraction (DELIVER) study was designed to evaluate the efficacy and safety of dapagliflozin in patients with HFpEF, regardless of whether the enrolled patients had diabetes [35]. The DELIVER study randomized 6263 patients (2747 women) with HF and an LVEF ≥ 40% to placebo or dapagliflozin administered at a dose of 10 mg once daily in addition to standard care medications [35]. The composite occurrence of worsening HF or cardiovascular death was the primary outcome. The primary endpoint appeared in 512 of 3131 patients (16.3%) in the dapagliflozin group and 610 of 3132 patients 21.2% in the placebo group over 2.8 years (hazard ratio, 0.82; 95% confidence interval [CI], 0.73 to 0.92; *p* < 0.001). In the dapagliflozin group, 368 patients (10.0%) experienced a first episode of worsening HF, compared to 455 patients (13.7%) in the placebo group (hazard ratio, 0.79; 95% CI, 0.69 to 0.91; *p* < 0.001). Cardiovascular death occurred in 231 patients (7.4%) of in the dapagliflozin group, whereas 261 patients (8.3%) in the placebo group (hazard ratio, 0.88; 95% CI, 0.74 to 1.05; *p* < 0.001). Regardless of EF or T2DM, dapagliflozin reduced the risk of HF deterioration and death from cardiovascular events compared to the placebo group. In terms of safety, 1361 patients (43.5%) in the dapagliflozin and 1432 patients (45.5%) in the placebo group developed serious adverse events, including death. The rate of study discontinuation due to any adverse events over 2.8 years was similar, 5.8%, in both groups.

## 4. Discussion

This is the first narrative review to examine the effect of dapagliflozin on exercise capacity and adverse cardiac events in patients with HF with or without T2DM. HF is an extremely incapacitating affliction affecting millions of people worldwide. There is still a significant unmet need for HF prevention through early detection and treatment of people who are symptomatic or at high risk of developing adverse events related to HF. The American College of Cardiology, the European Society of Cardiology, and the UK National Institute for Health and Care Excellence have recommended dapagliflozin in the treatment of patients with HFrEF [36,37,38]. The recommended dose of dapagliflozin is 10 mg once a day with or without food for patients with HF who have no severe liver dysfunction or renal dialysis.

Dapagliflozin has been associated with a lower risk of MACEs in patients with HF compared to placebo, regardless of the presence or absence of T2DM. The DEFINE-HF, DAPA-HF, and DECLARE–TIMI 58 trials demonstrated the cardiovascular benefits of dapagliflozin in patients with HFrEF, including reduced hospitalizations and death related to HF [8,29,33]. A recent meta-analysis looked into the effect of dapagliflozin on cardiovascular events as reported in 21 clinical studies [35]. This meta-analysis included a total of 9339 patients: 5936 patients received dapagliflozin (2.5–10 mg) and 3403 received a placebo. Remarkably, dapagliflozin administration led to a significantly decreased rate of hospitalization related to HF in patients with HFrEF compared to the control group. 

Similarly, the DELIVER trial demonstrated the beneficial effects of dapagliflozin in patients with HFpEF, including reduced worsening HF and cardiovascular death [34]. The beneficial effect of dapagliflozin on exercise capacity in patients with HF remains controversial. However, patients with HF have reported functional outcome measures that demonstrate a positive effect. The controversy between the trials is due to the heterogeneity of the sample recruited and the outcomes used to measure exercise capacity in these clinical trials. The DAPA-VO_2_ trial was designed primarily to evaluate the effect of dapagliflozin on exercise capacity and showed a positive effect on peak VO_2_, which is the gold standard for measuring cardiopulmonary exercise capacity [25]. This improvement may not be solely related to dapagliflozin, and other contributing factors, including weight loss associated with dapagliflozin may have overestimated the effect on peak VO_2_. Although there are several data available to show improvement in exercise capacity using subjective exercise testing measures, there are not enough data to show improvement in objective measures, and no conclusion has yet been reached. Furthermore, well-controlled studies using objective exercise testing are needed on patients with HF without comorbidities that affect exercise capacity.

In conclusion, a daily dose of 10 mg dapagliflozin along with standard care reduces the deterioration associated with HF and cardiac mortality in patients with HF, regardless of the presence or absence of T2DM. In addition, dapagliflozin has the potential to improve exercise capacity and requires further robust clinical studies with a good representative sample without comorbidities associated with exercise impairment. 

## Figures and Tables

**Figure 1 healthcare-10-02133-f001:**
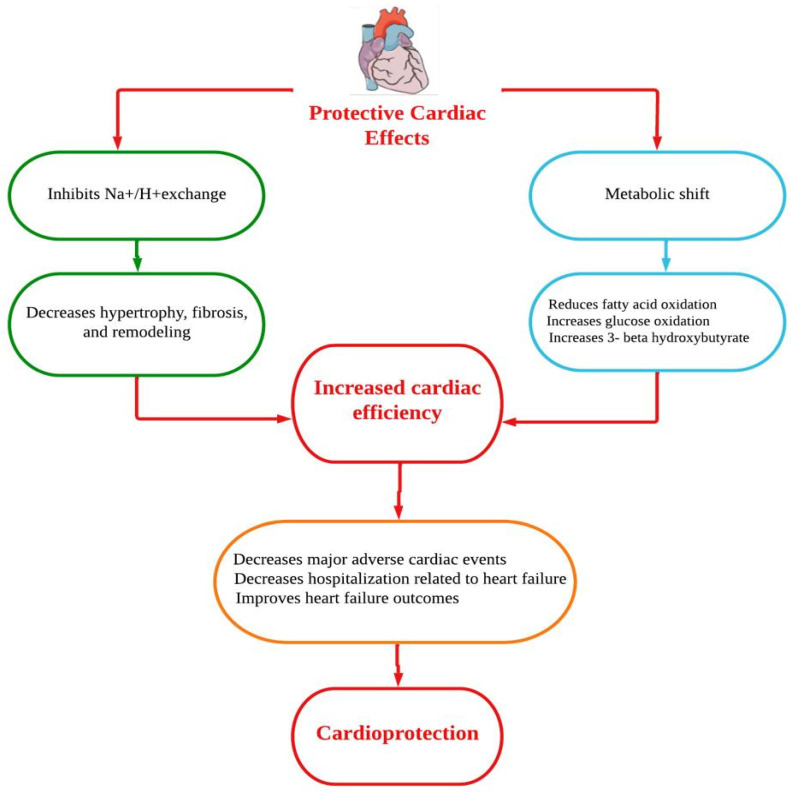
Putative cardioprotective mechanisms of dapagliflozin.

**Table 1 healthcare-10-02133-t001:** List of clinical trials evaluating the effect of dapagliflozin on exercise capacity in patients with heart failure.

Study Title	Treatment Condition	Study Type	Intervention	Trial Number
Dapagliflozin Effects on Biomarkers, Symptoms, and Functional Status in Patients with HF with Reduced Ejection Fraction (DEFINE-HF) [8]	HFrEF	Interventional	Dapagliflozin vs. placebo	NCT02653482
Dapagliflozin Effect on Exercise Capacity Using a 6-min Walk Test in Patients with Heart Failure with Reduced Ejection Fraction (DETERMINE) [24]	HFpEF and HFrEF	Interventional	Dapagliflozin vs. placebo	NCT03877237
Short-term effects of dapagliflozin on maximal functional capacity in heart failure with reduced ejection fraction (DAPA-VO_2_) [25]	HFrEF	Interventional	Dapagliflozin vs. placebo	NCT04197635
Effects of Dapagliflozin on Symptoms, Function, and Quality of Life in Patients With Heart Failure and Reduced Ejection Fraction [26]	HFrEF	Interventional	Dapagliflozin vs. placebo	NCT03036124
The SGLT2 inhibitor dapagliflozin in heart failure with preserved ejection fraction: a multicenter randomized trial [27]	HFpEF	Interventional	Dapagliflozin vs. placebo	NCT03030235
Dapagliflozin Effect on Exercise Capacity Using a 6-min Walk Test in Patients with Heart Failure with Reduced Ejection Fraction (DETERMINE) [28]	HFpEF	Interventional	Dapagliflozin vs. placebo	NCT03877224

Abbreviations: HFrEF, heart failure with reduced ejection fraction; HFpEF, heart failure with preserved ejection fraction.

**Table 2 healthcare-10-02133-t002:** List of clinical trials evaluating the effect of dapagliflozin on cardiovascular events in patients with heart failure.

Study Title	Treatment Condition	Study Type	Intervention	Trial Number
Efficacy and safety of dapagliflozin in men and women with heart failure with reduced ejection fraction [29]	HFrEF	Interventional	Dapagliflozin vs. placebo	NCT03036124
Effect of Dapagliflozin on Outpatient Worsening of Patients With Heart Failure and Reduced Ejection Fraction [30]	HFrEF	Interventional	Dapagliflozin vs. placebo	NCT03036124
Extrapolating Long-term Event-Free and Overall Survival With Dapagliflozin in Patients With Heart Failure and Reduced Ejection Fraction [31]	HFrEF	Interventional	Dapagliflozin vs. placebo	NCT03036124
Effect of Dapagliflozin on Worsening Heart Failure and Cardiovascular Death in Patients With Heart Failure With and Without Diabetes [32]	HFrEF	Interventional	Dapagliflozin vs. placebo	NCT03036124
Dapagliflozin Effect on Cardiovascular Events–Thrombolysis in Myocardial Infarction 58 (DECLARE–TIMI 58) [33]	HFrEF	Interventional	Dapagliflozin vs. placebo	NCT01730534
Dapagliflozin in Heart Failure with Mildly Reduced or Preserved Ejection Fraction (DELIVER) [34]	HFpEF	Interventional	Dapagliflozin vs. placebo	NCT03619213

## Data Availability

The data presented in this study are available on request from the corresponding author.

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
