# Peer review of "Effect of Dapagliflozin on Exercise Capacity and Cardiovascular Risk in Patients with Heart Failure"

_healthcare, 2022, doi:10.3390/healthcare10112133_

Round 1

Reviewer 1 Report (Previous Reviewer 2)

Reviewer comments and suggestions

The authors in this study want to explore the effect of dapagliflozin that was a new drug on exercise capacity and CV and therefore their goal and objective of this review is to summarize the effect of dapagliflozin on exercise capacity and cardiovascular risk in patients with HF.

I listed below a few suggestions to be incorporated in the revised version of the manuscript

  1. Introduction second paragraph first two lines need references to validate the points
  2. In the same paragraph the author written recent studies means author have to cite two or more reference here
  3. Third paragraph of the introduction section The author needs to define the exercise capacity for the general reader of your paper
  4. “The composite occurrenceof worsening HF or cardiovascular death was” please check the line 
  5. What does it indicate “the primary outcome. The primary endpoint appeared in 16.3% of patients in the dapagliflozin group and 21.2 % in the placebo group over 18.2 months “
  6. Comments for table 2 If possible you can add more trail number if there were some studies
  7. Prepare another para for the conclusion and try to add more points about using this drug as limitations that you pointed out from the trial.

Author Response

Comments of Reviewer 1

The authors in this study want to explore the effect of dapagliflozin that was a new drug on exercise capacity and CV and therefore their goal and objective of this review is to summarize the effect of dapagliflozin on exercise capacity and cardiovascular risk in patients with HF.

I listed below a few suggestions to be incorporated in the revised version of the manuscript

  1. Introduction second paragraph first two lines need references to validate the points

Response: References have been added.

  1. In the same paragraph the author written recent studies means author have to cite two or more reference here

Response: More reference have been added.

  1. Third paragraph of the introduction section, the author needs to define the exercise capacity for the general reader of your paper

Response: The definition of exercise capacity is added.

  1. “The composite occurrence of worsening HF or cardiovascular death was” please check the line

Response: The line is re-written.

  1. What does it indicate “the primary outcome. The primary endpoint appeared in 16.3% of patients in the dapagliflozin group and 21.2 % in the placebo group over 18.2 months “

Response: The composite occurrence of worsening HF or cardiovascular death was the primary outcome.

  1. Comments for table 2 If possible you can add more trail number if there were some studies

Response: A new recent trail has been added.

  1. Prepare another para for the conclusion and try to add more points about using this drug as limitations that you pointed out from the trial.

Response: The conclusion is re-written to reflect the discussion and convey the message of this review.

Reviewer 2 Report (Previous Reviewer 3)

I am very pleased to have been involved in this paper once again and to know that the authors considered my feedback.

I have had the opportunity to read this paper over and over again. Although the authors seem to evaluate the relationship of dapagliflozin to improved exercise capacity favorably, their arguments do not seem to be based on clear evidence: for all of the lengthy explanations of the importance of the 6-minute walk test early in chapter 2, only one study has objectively shown a benefit (the PRESERVED-HF). This is the same for the DAPA-VO2 study, which demonstrated an improvement in PeakVo2. Furthermore, the improvement in Peak VO2 shown in this study is also not as clinically significant as it could be. In addition, since the Peak VO2 in this study was a relative value divided by body weight, it is possible that the benefit of weight loss with SGLT2 inhibitors may have overestimated their effect. In summary, it is reasonable to conclude that, while there are several data available to show improvement in subjective measures, there are not enough data to show improvement in objective measures, and no conclusion has yet been reached. The study should have included any studies of other SGLT2 inhibitors, including empagliflozin, but this would not differentiate this study from other reviews that have already been published, thereby limiting this study.

If the first line on page 6 states that the drug is effective in heart failure regardless of left ventricular ejection fraction, I think the authors should mention not only the DPA-HF study but also the DELIVER study (PMID: 36027570).

Minor corrections

Despite numerous points, there are still many incorrect terminologies.

The English terminology should be consistent, including "randomize" and "randomise".

Page 6, line 1, what is a dapagliflozin inhibitor?

On page 6, line 11, put a space in "occurrenceof".

Page 8, last line, insert a space for "representativesample".

Table 1, "vs." should be consistent.

Page 5, line 6, what is " MED"?

Author Response

Comments of Reviewer 2

I am very pleased to have been involved in this paper once again and to know that the authors considered my feedback.

I have had the opportunity to read this paper over and over again. Although the authors seem to evaluate the relationship of dapagliflozin to improved exercise capacity favorably, their arguments do not seem to be based on clear evidence: for all of the lengthy explanations of the importance of the 6-minute walk test early in chapter 2, only one study has objectively shown a benefit (the PRESERVED-HF). This is the same for the DAPA-VO2 study, which demonstrated an improvement in PeakVo2. Furthermore, the improvement in Peak VO2 shown in this study is also not as clinically significant as it could be. In addition, since the Peak VO2 in this study was a relative value divided by body weight, it is possible that the benefit of weight loss with SGLT2 inhibitors may have overestimated their effect. In summary, it is reasonable to conclude that, while there are several data available to show improvement in subjective measures, there are not enough data to show improvement in objective measures, and no conclusion has yet been reached. The study should have included any studies of other SGLT2 inhibitors, including empagliflozin, but this would not differentiate this study from other reviews that have already been published, thereby limiting this study.

Response: Thank you for your valuable time and comments, and they are really have tremendously improved the review. We agree that there are no enough studies supporting the effect of DAPA on exercise capacity using objective exercise testing, and therefore this has been clearly mentioned in the discussion.

If the first line on page 6 states that the drug is effective in heart failure regardless of left ventricular ejection fraction, I think the authors should mention not only the DPA-HF study but also the DELIVER study (PMID: 36027570).

 Response: The DELIVER study has been added.

Minor corrections

Despite numerous points, there are still many incorrect terminologies.

Response: The manuscript has been proofread.

The English terminology should be consistent, including "randomize" and "randomise".

Response: The manuscript has been proofread.

Page 6, line 1, what is a dapagliflozin inhibitor?

Response: It has been corrected by deleting the word inhibitor.

On page 6, line 11, put a space in "occurrenceof".

Response: A space has been inserted.

Page 8, last line, insert a space for "representativesample".

Response: A space has been inserted.

Table 1, "vs." should be consistent.

Response: The “vs.” is kept consistent throughout the table 1.

Page 5, line 6, what is " MED"?

Response: It is meant to be the median, and now it is replaced by median.

Round 2

Reviewer 2 Report (Previous Reviewer 3)

I believe the authors have responded appropriately to my comments and revised the manuscript. There are minor revisions shown below, but this process was not my primary duty. I recommend that the authors revisit the manuscript again and correct any minor errors. This should be done at the authors' own responsibility.

Page 5, line 50, the "in" following the group is unnecessary.

Page 7, line 23, what is (5.8%)? Does it mean "similar"?

Author Response

Dear Reviewer,

Thank you for your feedback. We have corrected the changes required.

Best regards,

I believe the authors have responded appropriately to my comments and revised the manuscript. There are minor revisions shown below, but this process was not my primary duty. I recommend that the authors revisit the manuscript again and correct any minor errors. This should be done at the authors' own responsibility.

Page 5, line 50, the "in" following the group is unnecessary.

Response: Corrected

Page 7, line 23, what is (5.8%)? Does it mean "similar"?

Response: The rate of any adverse events was similar, 5.8%, in both groups.

This manuscript is a resubmission of an earlier submission. The following is a list of the peer review reports and author responses from that submission.

Round 1

Reviewer 1 Report

The Manuscript entitled: Effect of Dapagliflozin on exercise capacity and cardiovascular risk in patients with heart failure has been reviewed. The goal of this review is to highlight the effect of dapagliflozin on exercise capacity 54 and cardiovascular risk in individuals with HF, based on the available evidence. The authors have covered the following points in a very good way:

1. The question/objective to be dealt with

2. Select the best researchs to respond to the main question/objective

3. To conclude that Dapagliflozin is clearly associated with a statistically significant improvement in exercise capacity, and reduction in HF hospitalization rates, major adverse events, cardiovascular death, and all-cause mortality in patients with both heart failure.

I have no comments.

Reviewer 2 Report

Reviewer comments and suggestions 

The primary objective of this review was to summarize the effect of dapagliflozin on exercise capacity and cardiovascular risk in patients with Heart failure (HF). 

After going through the manuscript, I am not convinced with the structure they follow. The authors did not mention any study design on how they conducted the review. Better they need to follow some criteria for writing a review. What novelty they presented in the review was missing. Therefore I am rejecting the manuscript. For the improvement of the manuscript, I am suggesting a few comments. 

  1. Line 32 Few studies are mentioned by the authors so cite a more relevant manuscript
  2. Line 42-42 What would be the reason for this
  3. I thought there would be some meta-analysis or any collection of data on how the authors proposed the study. It would be nice if they can extend the study by increasing tables and mechanism-related figures.  
  4. Line 72-73 The authors need to modify the sentences “The mechanisms through which dapagliflozin protects the heart and circulatory system is sophisticated and, in some ways, unknown”
  5. Line 74-75 The points needed references
  6. Line 82-92 It would be nice if the authors used some figures instead of writing a paragraph
  7. Line 110 -111 please modify the sentence
  8. Line 138-139 is these was study points or the authors predicted
  9. Line 225-242 as previously suggested to prepare tables or meta-analysis. The structure of the manuscript is really poor

Reviewer 3 Report

This review systematically discusses the effects of the dapagliflozin, a major SGLT2 inhibitor, on exercise capacity and cardiovascular outcomes, as demonstrated in several studies, including the DAPA-HF and DECLARE-TIMI trials. The improvement in outcomes with SGLT2 inhibitors does not correlate with improvement in physical function (PMID: 35600478), and a meta-analysis found that the drug improved quality of life but not 6-minute walking distance (PMID: 34653575). This is also the case for dapagliflozin, and I do not particularly question the authors' view.

Considering the history of reviews and mea-analyses that have already been reported, I think the originality of the significance of this new report is that it is limited to dapagliflozin, but I do not think that it is especially novel. As described below, there are also doubts about the references, and I would like the authors to consider resubmitting the manuscript after properly addressing them.

Lines 141-177: The DETERMINE trial cited by the authors is registered on Clinicaltrials.gov (NCT03877237 and NCT03877224, respectively) and results are reported briefly, but not linked as a published article on PubMed. Understanding the background factors of the patients enrolled is crucial to interpreting clinical outcome results, but no information other than age, gender, and race is provided; it is not possible to assess what might account for the differences in results between the DETERMINE-Preserved and PRESERVED-HF trials . Not only that, it must be said that it is very difficult to analogize generalizability from the results of the DETERMINE study.

If citations seem scarce, please refer to the following papers.

Palau P, Amiguet M, et al. Short-term effects of dapagliflozin on maximal functional capacity in heart failure with reduced ejection fraction (DAPA-VO2 ): a randomized clinical trial. Eur J Heart Fail. 2022 May 23. doi: 10.1002/ejhf.2560.

Minor corrections

Line 30: Spell out "T2DM".

Line 82: Shouldn't it be "inflammatory minimisation"?

Line 131: There is a mix of American and British English. Please review the entire document.

Line 146: Please correct the typographical error.

Line 179: The source of the citation appears to be a sub-study of DAPA-HF (PMID: 31736335), but it should be clearly indicated.

Line 192: The closing parentheses should be aligned with the rest of the form, as should lines 263 and 266.

Line 204: Is "of" necessary?

Round 2

Reviewer 2 Report

No more comments

Reviewer 3 Report

Thank you for responding to my peer review comments.

The authors' response is simple and I am not sure they have understood my comments properly. I leave it to the editor-in-chief to make the final decision on handling of DETERMINE trials based on unpublished data, as this is a matter of journal policy, but even so, I think it is unfair to present the results in the text without indicating the source. At the very least, the Clinicaltrials.com URL should be mentioned. Furthermore, the reference 24 that the authors cited as their source does not include the study in question. This is probably the last opportunity for me to comment on this issue, so I think it would be better to address this issue before publication.

The authors stated in the conclusion section that "Dapagliflozin is clearly associated with a statistically significant improvement in exercise capacity". This cannot be interpreted in the context of the text. How should the discrepancy between the results of cardiopulmonary exercise testing and the six-minute walk test be interpreted? Should the occurrence of myopathy due to empagliflozin, which I personally do not experience very often, be considered as not affecting the results of the cardiopulmonary exercise stress testing? Given that some patients with heart failure are unable to adequately perform the gait test, e.g. due to orthopaedic conditions, it is quite possible that the presence of such a subgroup may result in a neutral outcome in the gait test. Please present previous studies that is in line with the authors' views, rather than simply transcribing the results of each clinical study.

Line 41: "recent studies" is assumed to refer to reference 11, but the source should be identified.

Line 24: What is meant by 'assist cardiopulmonary resuscitation'?

Line 147: What does 'critiaa' mean?

Line 182: Is the DTERMINE-Preserved trial a substudy of DAPA-HF?

Line 241: It appears that the data from the source has been incorrectly transcribed. Does the placebo group have a longer survival?

Line 254: "the of" is unnecessary.

The mixing of American and British English has not been corrected. The rules for closing brackets are also not consistent. If the authors cannot address these errors, they should bring in a proper external proofreader.
